



# Quantifying daily NO$_x$ and CO$_2$ emissions from Wuhan using satellite observations from TROPOMI and OCO-2

Qianqian Zhang[1,2], K. Folkert Boersma[1,3], Bin Zhao[4], Henk Eskes[3], Cuihong Chen[5], Haotian Zheng[4], Xingying Zhang[2]

[1] Wageningen University, Environmental Science Group, Wageningen, the Netherlands
[2] Key Laboratory of Radiometric Calibration and Validation for Environmental Satellites, Innovation Center for Fengyun Meteorological Satellite (FYSIC), National Satellite Meteorological Center, China Meteorology Administration, Beijing, 100081, China
[3] Royal Netherlands Meteorological Institute, De Bilt, the Netherlands
[4] State Key Joint Laboratory of Environmental Simulation and Pollution Control, School of environment, Tsinghua University, Beijing 100084, China
[5] Satellite Application Center for Ecology and Environment, Ministry of Ecology and Environment of the People's Republic of China, Beijing, 100094, China

*Correspondence to*: K. Folkert Boersma, folkert.boersma@wur.nl, Qianqian Zhang, zhangqq@cma.gov.cn

**Abstract.** Quantification and control of NO$_x$ and CO$_2$ emissions are important across the world to limit adverse climate change. We present a new top-down method, an improved superposition column model to estimate day-to-day NO$_x$ and CO$_2$ emissions from the large city of Wuhan, China, located in a polluted background. The lasted released version 2.3.1 TROPOMI NO$_2$ columns and the version 10r of the OCO-2 observed CO$_2$ mixing ratio are employed. Our estimated NO$_x$ and CO$_2$ emissions from Wuhan are verified against bottom-up inventories with small deviations (< 3 %). Based on the estimated CO$_2$ emissions, we also predicted daily CO$_2$ column mixing ratio enhancements, which match well with OCO-2 observations (< 5 % bias, within ±0.3 ppm). We capture the day-to-day variation of NO$_x$ and CO$_2$ emissions from Wuhan in 2019–2020, which does not reveal a substantial 'weekend reduction' but does show a clear 'holiday reduction' in the NO$_x$ and CO$_2$ emissions. Our method also quantifies the abrupt decrease and slow rebound of NO$_x$ and CO$_2$ emissions due to the Wuhan lockdown in early 2020. This work demonstrates the improved superposition model to be a promising new tool for the quantification of city NO$_x$ and CO$_2$ emissions, allowing policy makers to gain real-time information into spatial-temporal emission patterns and the effectiveness of carbon and nitrogen regulation in urban environments.



## 1 Introduction

Fossil fuel combustion by power plants, industrial activities, transportation, and residential energy use sectors leads to emission of nitrogen oxides ($NO_x$ = NO + $NO_2$) as well as carbon dioxide ($CO_2$). Traditional bottom-up $NO_x$ and $CO_2$ emission estimates have a lag in time of several years, because it takes time to access and compile accurate information on energy consumption and the emission factors (Lamsal et al., 2011; Liu et al., 2020a).

For decades satellites have been continuously providing information of $NO_2$ distributions and trends with good quality, and satellite data is widely used to quantify $NO_x$ emissions and changes(Lamsal et al., 2010; Visser et al., 2019; Zhang et al., 2020; Zhang et al., 2021). Based on satellite retrieved $NO_2$ data, previous studies quantified long-term mean (monthly, yearly or multi-yearly) $NO_x$ emissions on global and regional scales(Lamsal et al., 2011; Visser et al., 2019). Beirle et al. (2011) analyzed downwind plumes of satellite $NO_2$ columns averaged on each wind direction, and then inferred $NO_x$ emissions from isolated large point sources and megacities. Inspired by this idea, Lorente et al. (2019) analyzed the increase of $NO_2$ along with the wind over the extensive pollution source of Paris. The build-up of $NO_2$ over the city observed from space, in combination with information on wind speed and direction allows to obtain day-by-day (sub-)urban $NO_x$ emission estimates and lifetimes as long as the city is under a clear sky and winds are relatively constant in time. This approach does not need burdensome inverse modelling computations and opens possibilities for rapid and direct monitoring of $NO_x$ emissions from space.

In contrast to $NO_x$, it is challenging to infer accurate localized anthropogenic $CO_2$ emissions from satellite $CO_2$ retrievals. One reason is that the background $CO_2$ concentration is orders of magnitude higher than the enhancement caused by anthropogenic emissions, reflecting the long atmospheric lifetime of $CO_2$ (Reuter et al., 2014; 2019). Another reason is that the spatial and temporal coverage of current $CO_2$ sensors is too sparse to allow substantial averaging of noisy signals by revisiting of scenes, precluding detailed $CO_2$ emission estimation(Zheng et al., 2020a; Liu et al., 2020a). Using satellite $NO_2$ measurements to estimate anthropogenic $NO_x$ emissions as the basis to infer anthropogenic $CO_2$ emission has been proposed in several studies(Reuter et al., 2019; Liu et al., 2020a; Berezin et al., 2013; Zheng et al., 2020a). However, to our knowledge there is no method that estimates day-to-day top-down $CO_2$ emission estimation on (sub-)city scale.

Here we revisit the method of Lorente et al. (2019) to improve our understanding of its potential and limitations and extend it to estimate city-scale daily $NO_x$ and $CO_2$ emissions. We present an improved superposition model that considers the build-up of pollution over a source area as in Lorente et al. (2019), as well as the decay of $NO_2$ downwind of the source, but now also accounts for changes in the background $NO_2$ pollution along wind direction. The background $NO_2$ pollution was considered to remain constant in Lorente et al. (2019) for Paris, which is not surrounded by significant surface sources of $NO_x$ pollution. Here we apply our improved method on a highly polluted urban area, the megacity of Wuhan in Hubei Province of China, which, other than the relatively isolated city of Paris, is located in a polluted background with many surrounding surface pollution sources that potentially interfere with the build-up and decay of the $NO_2$ plume from Wuhan. Using this improved superposition model, together with the bottom-up information on $CO_2$-to-$NO_x$ emission ratio, we infer





$NO_x$ and predict $CO_2$ emissions on a day-by-day basis over a full year, from September 2019 to August 2020, and analyze the variation in emissions and $NO_x$ chemical lifetime from day to day. Of particular interest are the reductions and subsequent rebound of $NO_x$ and $CO_2$ emissions associated with the COVID-19 lockdown measures in Wuhan, which have been reported in other studies, and serve here as a useful check on the robustness of our method.

## 2 Data and Material

**2.1 Satellite data**

In this study, we use the newly released level-2, version 2.3.1 of the S-5P TROPOMI data (TROPOMI-v2.3.1) between September 2019 to August 2020. The S-5P (Sentinel-5 Precursor) satellite was launched in October 2017, and the TROPOMI (TROPOspheric Monitoring Instrument) on board provides tropospheric $NO_2$ columns with a unprecedented horizontal resolution up to 5.5km × 3.5km (as of 6 August 2019) and high signal-to-noise ratio(Griffin et al., 2019; Van

Geffen et al., 2020). The v2.3.1 dataset is provided by S5P-PAL (Eskes et al., 2021), and is dedicated to support the research on the impact of the COVID lockdown on air quality. Compared to the earlier version, this dataset has 10–40 % higher tropospheric $NO_2$ columns over polluted scenes due to the improved cloud retrieval and other algorithm updates (Van Geffen et al., 2022; Riess et al., 2022). Over Wuhan we find an average increase of about 25 %, but the difference between the two versions changes spatially and temporally (Fig. S2). According to Fig. S2, the increase in v2.3.1 is much higher over

polluted area (city center) and polluted period (9 September and 3 October 2019). Improved (residual) cloud pressures correct the low bias of v1.x data compared to OMI and ground-based measurements over east China (Wang et al., 2020; Liu et al., 2020b). In addition, an improved treatment for the surface albedo increases the columns for cloud-free scenes (Van Geffen et al., 2022). For comparison (and to assess the impact of retrieval improvements on $NO_x$ emission estimates), we also use version 1.3 data (TROPOMI-v1.3) for 2019 to derive $NO_x$ emissions from Wuhan. Since previous studies have

pointed out a low bias in the v1.x TROPOMI retrieval, especially over China (Griffin et al., 2019; Wang et al., 2020; Liu et al., 2020b), we scaled up the v1.3 $NO_2$ columns by a factor of 1.6 to correct for this known -40 % bias in TROPOMI $NO_2$ data as reported by Liu et al. (2020b).

    We sampled the TROPOMI $NO_2$ columns into 0.05° lat × 0.05° lon grid cells (~ 6 × 6 km$^2$). To assure good data quality, we filtered out the data with cloud radiance fractions greater than 0.5 (qa_value > 0.75), and obtain 81 clear-sky days with

full TROPOMI $NO_2$ coverage over the Wuhan region in one full year.

    The column-averaged dry air mole fraction of $CO_2$ (XCO$_2$) data provided by the Orbiting Carbon Observatory-2 (OCO-2) are also employed to verify the derived $CO_2$ emission inventory for Wuhan. We use the version 10r of the bias-corrected XCO$_2$ product (Gunson M and Eldering, 2020). The v10 OCO-2 XCO$_2$ product has high accuracy with single sounding precision of ~0.8 ppm over land and ~0.5 ppm over water, and RMS biases of 0.5–0.7 ppm over both land and water (Odell

et al., 2021).





## 2.2 Bottom-up emission information

Bottom-up $NO_x$ and $CO_2$ emission inventories are used to provide the first-guess of $NO_x$ emission spatial pattern (for $NO_x$, in the Supplement, Text S1 and Fig. S1) and to verify the top-down emissions. We use the Air Benefit and Attainment and Cost Assessment System Emission Inventory (ABACAS) (Zhao et al., 2013; Zhao et al., 2018; Zheng et al., 2019) , which

provides $NO_x$ and $CO_2$ emissions for the year 2019. The Multi-resolution Emission Inventory (MEIC) (Li et al., 2017) $NO_x$ emissions for 2017 are also used.

## 2.3 Other input data

Besides the satellite data and bottom-up emission inventories, a set of other parameters are used to input into our improved superposition model. They include the hydroxyl radical (OH) concentration, the loss rate ($k$) of $NO_x$ in the atmosphere, the

$[NO_x]\big/[NO_2]$ ratio, and the wind field. The first three are from the GEOS-Chem chemical transport model, and the wind field is from ERA5, the fifth generation ECMWF atmospheric reanalysis of the global climate (Hersbach et al., 2020). Detailed information of these data is seen in the supplement, Text S2.

## 2.4 NO₂ pattern fits: estimation of lifetime and emission

To ensure that the whole area of Wuhan is included, we determine our study domain as a circular region centered at 114°

E, 30.7° N, with a diameter of ~186 km. It includes the whole area of Wuhan and the small city of Ezhou to the east of Wuhan, the southwest part of Huanggang and east part of Xiaogan (Fig. S3). We also do a sensitivity test to narrow the study area down to within the Third Ring Road of Wuhan to check the robustness of our model to the area size of study domain (Fig. S3). For each day, we converted the two-dimensional $NO_2$ column map over the domain to a one-dimensional line density along the wind direction (Text S3) (Beirle, 2011; Lorente et al., 2019). $NO_x$ emissions and lifetimes can be estimated

through the fitting of the $NO_2$ line density over the domain.

Lorente et al. (2019) presented a superposition model based on a column model (Jacob, 1999) to simulate $NO_2$ line density over Paris. They considered the build-up of $NO_2$ caused by spatially varying $NO_x$ emissions from each cell and used the $NO_2$ line density value at the upwind end of the city to represent the background value, which they assumed to be constant over the city. This appears plausible if the background value would mostly represent free tropospheric $NO_2$ which has a longer

lifetime than $NO_2$ in the oxidizing polluted boundary layer and varies smoothly according to models. Our method to simulate the $NO_2$ line density over the city is also based on the column model (Jacob, 1999), but differs from that of Lorente et al. (2019) in considering the background $NO_2$ value. Each cell along the wind direction is treated separately as a column model. $NO_x$ emissions from the current cell contribute to the total line density through the build-up of $NO_2$ density within the cell and exponential decay of $NO_2$ downwind of the cell (Eq. (1)). It doesn't contribute to the upwind cells (Eq. (2)).




$\quad N_i(x) = \frac{E_i}{k}\left(1 - e^{-kL/u}\right) \times e^{-k(x-x_i)/u} \times \frac{[NO_2]}{[NO_x]} \quad for\ x > x_i \quad ,$ (1)

$\quad N_i(x) = 0 \qquad\qquad\qquad\qquad\qquad for\ x \le x_i \ ,$ (2)

where $N_i$ represents the NO$_2$ line density ( molec/cm) contributed from $E_i$ in cell $i$, $L$ is the length of each cell, i.e. 600000 cm; $k$ is the loss rate (s$^{-1}$) of NO$_x$ at 13:00 local time ( $k = \frac{k'[OH]}{[NO_x]/[NO_2]}$ ); and $u$ denotes the NO$_2$-density-weighted mean wind speed in unit of cm/s within planet boundary. We add up the contributions from each cell and the background value to model the

overall NO$_2$ line density:

$\quad N(x) = \sum_{i=1}^{n} N_i(x) + b + \alpha x \ ,$ (3)

here, $b$ represents the starting background value, equivalent to the mean NO$_2$ line density within the e-folding distance(Liu et al., 2016) upwind of $x$=0. $\alpha$ denotes the linear change of background value with distance along wind, and represents the chemical decay of background NO$_2$ flowing into the polluted boundary layer over the city.

$\quad$ We fit the terms that drive $N(x)$ (i.e. $E_i$, $k$ and $\alpha$) with the fixed $L$, $u$ and $[NO_x]/[NO_2]$ from external data, via a least-squares minimization to the TROPOMI observed line density $N_{TROPOMI}(x)$. For each day, we run the model 20 times randomly choosing OH concentration within the ±20% interval of GEOS-Chem simulated OH concentration. The mean value from the 5 sets of parameters $E_i$, $k$ and $\alpha$ that best explain the observations over the city is the answer we are seeking for. The parameter that describes the decay of upwind NO$_2$ over the city, the $\alpha$ value, is determined by the difference of NO$_2$

line density between the end and start point of the study domain, $\alpha = \frac{(N_{31} - N_1)}{30L}$, and we allow it to change between ±$\alpha$ in the fitting procedure. For the 50 days on average, the $\alpha$ value is $(-0.006 \pm 0.008) \times 10^{-22} molec/cm^2$. The $\alpha$ value being negative reflects the decay of upwind NO$_2$ pollution along the wind.

$\quad$ The assumption of a linearly decreasing NO$_2$ background is relevant under conditions when the city is in a polluted background. It accounts for decay of upwind NO$_2$ pollution arriving at the city when transported over and downwind of the

city. In reality, upwind NO$_2$ pollution mixes in with the freshly emitted NO$_x$ and is then subject to chemical decay (with non-linearities due to turbulent mixing and spatial heterogeneity in emissions). We acknowledge that our linear decrease of background NO$_2$ pollution is a severe simplification, but as shown in Fig. 1, compared to fitting results with a constant background value, we obtain a better correlation (up to 25%) between fitted and observed NO$_2$ line densities when fitting with a linearly changing background value.






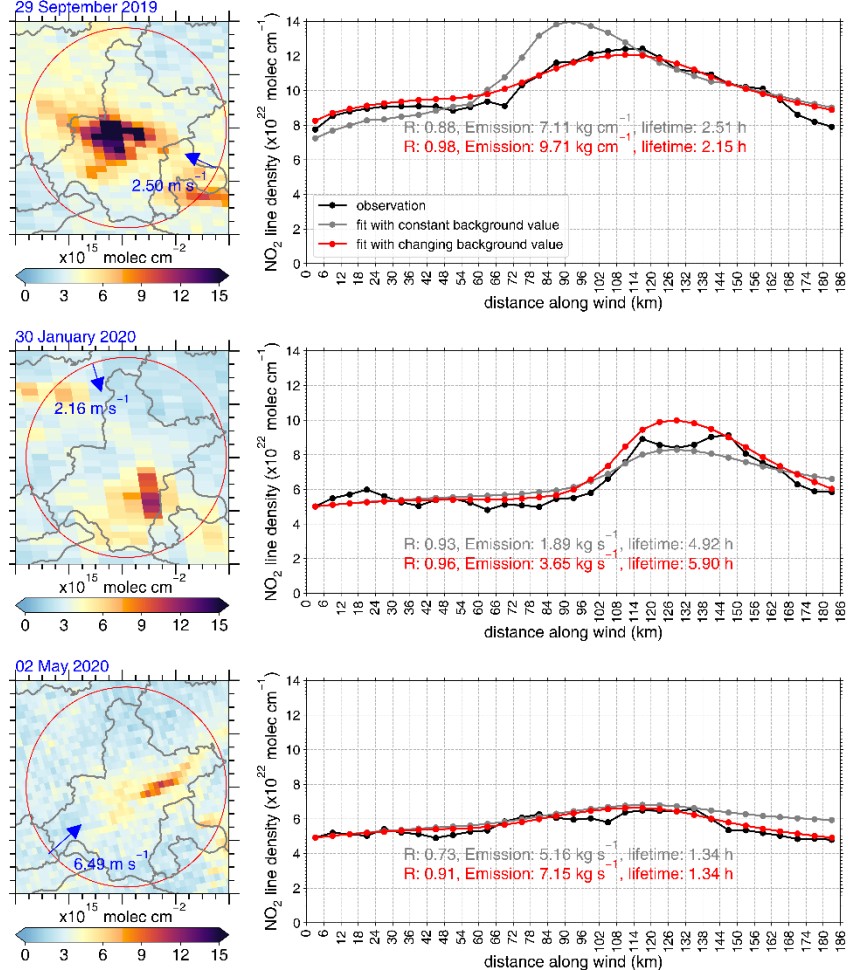

**Figure 1: Tropospheric NO₂ columns over Wuhan on 29 September 2019, 30 January 2020 and 2 May 2020 (left, from top to bottom), the red circle inside each map defines the study domain. The corresponding NO₂ line densities along wind within the study domain are given in the right panel.**

**2.5 CO₂ emission estimation**

City-scale CO₂ emissions are estimated through Eq. (4):

$$E_{CO_2} = E_{NO_x} \times Ratio_{CO_2-to-NO_x} , \tag{4}$$

The anthropogenic CO₂-to-NO$_x$ emission ratio is provided by the ABACAS inventory, and amounts to ~591 gCO₂/gNO$_x$ emitted from our study domain for the year 2019. In 2020, emissions from the transport sector have substantially decreased

due to the lockdown measurements(Huang et al., 2021; Zheng et al., 2020c). The stronger decrease in transport NO$_x$ emissions relative to decreases from other sectors are predicted to have led to an increase in the CO₂-to-NO$_x$ emission ratio,



for this ratio is lowest in the transport sector (Zheng et al., 2020c). The monthly $CO_2$-to-$NO_x$ emission ratio for Wuhan were calculated based on recent reports on sectoral $NO_x$ emission in 2020 from Hubei Province (Zheng et al., 2021a). We then further calculated daily $CO_2$-to-$NO_x$ emission ratio based on the monthly, daily and diurnal variation of $CO_2$ and $NO_x$

emissions (Fig. S5). The final daily $CO_2$-to-$NO_x$ emission ratio for the study period displayed in Table S1 indeed shows increases in the $CO_2$-to-$NO_x$ emission ratio of up to 20 % during the lockdown period in 2020.

### 2.6 uncertainty in $NO_x$ and $CO_2$ emission estimation

Uncertainties in quantifying $NO_x$ and $CO_2$ emissions contain the systematic error in the TROPOMI $NO_2$ retrieval, bias in the assumed a priori OH concentration, $NO_x/NO_2$ ratio, $CO_2$-to-$NO_x$ emission ratio, uncertainties in wind fields and the area of

study domain. The $NO_2$ column dataset S5P-PAL corrected the low bias in TROPOMI (v1.x) tropospheric $NO_2$ column over Eastern China by 15–100 % (Van Geffen et al., 2022). The CTMs have difficulty in simulating accurate OH concentration, but for > 90 % of the days, our fitted OH concentrations fall in ±20 % range around GEOS-Chem simulation, so the uncertainty in OH concentration is likely on the order of ±20 %. The difference between model simulated and observed $NO_x/NO_2$ ratio is less than 10 %, so we give an uncertainty in $NO_x/NO_2$ ratio of ±10 %. Uncertainty in $CO_2$-to-$NO_x$ emission

ratio comes from the errors in sectoral $NO_x$ and $CO_2$ emissions, and we calculated that the uncertainty in $CO_2$-to-$NO_x$ emission ratio is ±30 %. We have narrowed down our study domain to check the sensitivity of our method to the area of study domain (see Fig. S4). The results demonstrate that when the study domain is narrowed down to 84 km diameter, the change in fitted $NO_x$ lifetime and $NO_x$ emission is less than ±15%. We use the $NO_2$-column-weighted mean instead of the arithmetic mean value to get the boundary layer mean wind speed to minimize the error in wind field, but there may remain

±20 % uncertainty in the ERA5 reanalysis data. Considering that all these parameters are independent from each other, we use the root mean square sum of the contributions to represent the overall uncertainty estimation, which we quantify for $NO_x$ lifetime and emission on a single day at ~37 %, and for $CO_2$ emission at ~48 %.

## 3 Results and discussion

### 3.1 $NO_x$ Lifetimes and emissions

We display the calculated $NO_x$ lifetime and $NO_x$ emission for each clear-sky day during the study period in Table S1. Fitted planetary boundary layer mean OH concentration over the region for each day is presented in Fig. 2. For 90 % of the days, our model fitted OH concentrations which fall into the intervals of 0.8~1.2× the GEOS-Chem model values. There are only 5 days on which we had to impose a change in OH concentrations of more than 30 % relative to the GEOS-Chem simulation to obtain realistic fitting results.

190 .





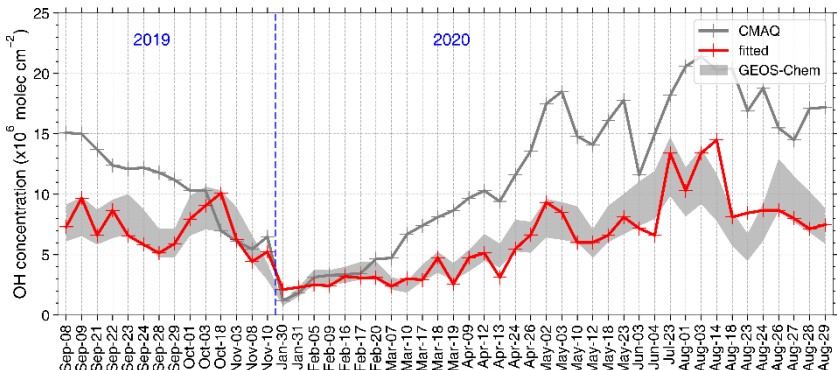

**Figure 2: Boundary layer mean OH concentration over our study domain. The gray shade represents 0.8–1.2 times GEOS-Chem simulated OH concentration.**

We estimate that seasonal mean noontime $NO_x$ lifetime over Wuhan and adjacent region is 4.8±0.8 h for winter, 2.8±1.3 h for spring, 1.4±0.3 h for summer and 1.9±0.5 h for autumn. The results are lower than those calculated from GEOS-Chem simulation by Shah et al. (2020), with ~6 h in summer and >20 h in winter. This is because they calculated the 24-hour mean $NO_x$ lifetime and the loss rate of $NO_x$ is much higher around noon. $NO_x$ lifetime for Wuhan is also shorter than for Paris (Lorente et al., 2019), especially during winter, reflecting the higher radiation levels and temperature in Wuhan than in Paris. It should be noted that Liu et al. (2016) fitted a $NO_x$ lifetime of 2.6 h for Wuhan in warm season (May to September) for 2005–2013 mean, and our result for 2019-2020 is 1.7±0.4 h. One reason is that they calculated $NO_x$ lifetimes based on a long-term mean $NO_2$ distribution, and the coarser resolution of OMI data, both of which lead to spatial smoothing of $NO_2$ gradients and thus longer apparent $NO_x$ lifetimes (Qu, 2020). Another explanation is the increasing ozone concentrations in China in recent years (Li et al., 2020) which promote OH formation and thereby $NO_x$ loss reactions which shorten $NO_x$ lifetimes (Zara et al., 2021).





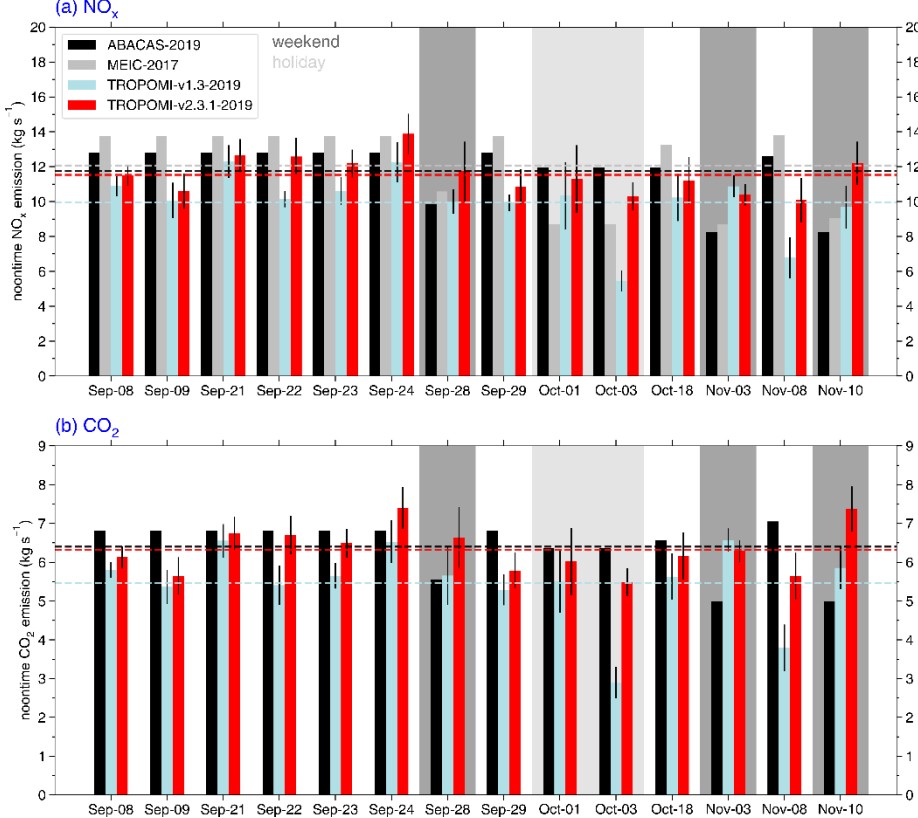

**Figure 3: Daily noontime (a) NO$_x$ and (b) CO$_2$ emission in Wuhan estimated from TROPOMI (red and blue bars, the error bars represent the standard deviation of the five best estimates for each day) and the bottom-up emission inventories ABACAS (black bars) for the year 2019 and MEIC (silver bars) for the year 2017. The dark and light grey shades represents weekends and holidays, respectively. Mean levels of each dataset are given as dashed lines with corresponding colors.**

The calculated noontime (13:00 local time) NO$_x$ emissions from Wuhan for 14 days (including 9 weekdays, 3 weekends and 2 holidays) between September and November 2019 are compared with those from the ABACAS (2019) and MEIC (2017) inventories. Overall, as presented in Fig. 3a, the rescaled TROPOMI-v1.3 estimated noontime NO$_x$ emissions are 13.6% lower than those from TROPOMI-v2.3.1. Compared to the bottom-up emission inventories, TROPOMI-v1.3-2019 NO$_x$ emissions are 15.1 % and 17.5 % lower than ABACAS-2019 and MEIC-2017, respectively. On the other hand, TROPOMI-v2.3.1-2019 NO$_x$ emissions are comparable to those from ABACAS-2019 (< 3 % difference), and ~5 % lower than MEIC-2017. That NO$_x$ emissions estimated from TROPOMI-v2.3.1 in 2019 are lower than MEIC-2017 likely reflects the fact that NO$_x$ emissions have decreased in 2019 relative to 2017 in response to Chinese emission controls. According to Wuhan Bureau of Statistics, NO$_x$ emissions have decreased 6.2 % between 2017 and 2019 (Statistics, 2019; Bauwens et al., 2020), close to the difference between TROPOMI-v2.3.1-2019 and MEIC-2017. TROPOMI-v2.3.1 NO$_2$ data generates more reliable NO$_x$ emissions from Wuhan in 2019 than the v1.3 data, even when the latter is scaled up by a factor of 1.6.



Different to the bottom-up inventories, our daily TROPOMI $NO_x$ emissions do not indicate the existence of a so-called 'weekend reduction effect', but do point out a distinct 'holiday reduction effect' in Wuhan $NO_x$ emissions. The bottom-up inventories suggest that weekend $NO_x$ emissions are 30 % reduced relative to weekdays. The TROPOMI estimation shows reductions in weekend $NO_x$ emission of < 3 %, while on the two days (1 and 3 October) of the National Holiday, $NO_x$ emissions are 8% lower than the workday mean. Surface $NO_2$ and $O_3$ observations from Beijing do not show a weekend effect (Zhao et al., 2019; Hua et al., 2021) either. Our TROPOMI $NO_x$ emissions show a similar spatial pattern as in ABACAS and MEIC (Fig. S1), with the highest emissions located in the city center of Wuhan. However, TROPOMI fitted a more smeared-out $NO_x$ emission pattern than ABACAS, due to the strong dependence of the bottom-up spatial distribution on population density, the difference in spatial resolution, and the decrease in $NO_x$ emission between 2017 and 2020 mainly occurring in the high-emission region.

### 3.2 $CO_2$ emissions and $XCO_2$ enhancements

We estimate noontime $CO_2$ emissions from Wuhan between September and November 2019 to be 6.32±0.60 s$^{-1}$, comparable to ABACAS-2019, of 6.40±0.70 t s$^{-1}$ (Fig. 3b). Based on the estimated daily $CO_2$ emissions, we further use the superposition column model to estimate daily $XCO_2$ enhancements, and validate them by OCO-2 observations. We successfully obtained two days between May 2018 (start time of TROPOMI-v2.3.1 $NO_2$ product) and December 2021 with simultaneous (both overpass at around 13:00 local time), co-located TROPOMI $NO_2$ and OCO-2 $CO_2$ observations over Wuhan: 15 September 2018 and 13 April 2020. We inferred total $CO_2$ emissions from Wuhan based on our TROPOMI-based $NO_x$ emissions and the ABACAS-predicted $CO_2$-to-$NO_x$ emission ratios on 15 September 2018 and 13 April 2020 to be 7.92±0.93 t s$^{-1}$ and 4.44±0.50 t s$^{-1}$ (the errors represent the standard deviation of the 5 best estimations for each day), respectively. We then scaled down the ABACAS 1×1 km$^2$ gridded $CO_2$ emissions to match 7.92 t s$^{-1}$ and 4.44 t s$^{-1}$, and then predict the $XCO_2$ enhancements using the top-down $CO_2$ emissions in combination with the superposition column model. It should be noted that to compare with the sparse distributed OCO-2 observations, we apply the superposition model on the $CO_2$ line density with 1km wide, while it is 186km wide for $NO_2$. Since that the column model doesn't take the dispersion of $NO_2$ or $CO_2$ into account, all dispersion will then be encapsulated within the domain when a line density covers cross-section as wide as 186 km. However, when the line density is only 1 km wide, the dispersion will move $CO_2$ out of this line, and we will discuss its influence on $CO_2$ enhancement estimation in the further below.

Neglecting chemical production and loss of $CO_2$ in the atmosphere, the superposition column model of $CO_2$ (Eq. 5) is simpler than that of $NO_x$:

$$N_{CO_2} = \frac{E_{CO_2}}{uL},$$

(5)





Here $N_{CO_2}$ is $CO_2$ density in unit of g m$^{-2}$, $E_{CO_2}$ denotes our top-down $CO_2$ emission (g s$^{-1}$), and $u$ and $L$ are the wind speed (m/s) and length of grid cell (6000 m). Then $N_{CO_2}$ (g m$^{-2}$) is converted to the column mixing ratio $XCO_2$ (ppm) to compare with the OCO-2 observation (Zheng et al., 2020a):

$$XCO_2 = N_{CO_2} \times \frac{M_{air}}{M_{CO_2}} \times \frac{g}{p-wg} \times 10^3 , \qquad (6)$$

in which $M_{air}$ and $M_{CO_2}$ are air and $CO_2$ molar mass of air and $CO_2$ (g mol$^{-1}$), $g$ is the gravitational acceleration (9.8 m/s$^2$), $p$ (Pa) and $w$ (kg m$^{-2}$) are surface pressure and total column water vapor, respectively.

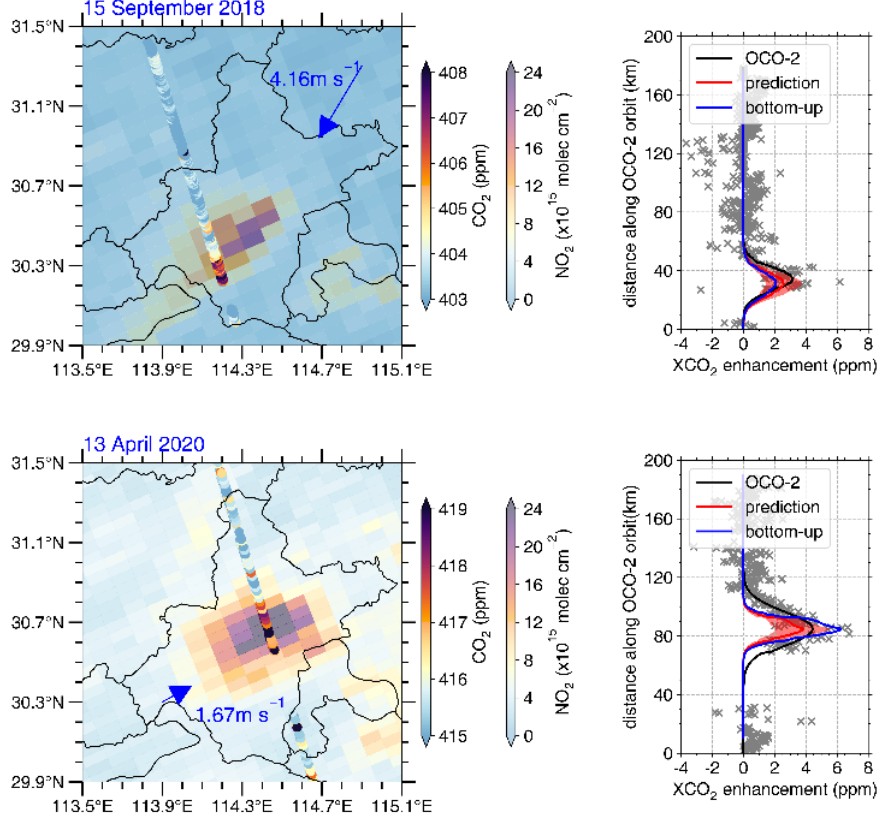

**Figure 4: Simultaneous, co-located TROPOMI NO₂ and OCO-2 CO₂ observations over Wuhan (left panel) on 15 September 2018 (top) and 13 April 2020 (bottom), wind speed and direction on each day are shown. The dry air mole fraction of CO₂ (XCO₂)**
**enhancements along the OCO-2 orbit are given for corresponding day (right panel). The gray xs and black lines represent the OCO-2 observation. The blue lines denote XCO₂ enhancement estimated with bottom-up emissions, and the red lines (shading represents the uncertainty interval) with CO₂ emission predicted in this study.**

We calculate the $XCO_2$ enhancement due our top-down $CO_2$ emissions on 15 September 2018 and 13 April 2020 and compare these with the enhancements observed by OCO-2. As shown in the right panels of Fig. 4, the superposition model
captures the spatial pattern of observed $XCO_2$ along the OCO-2 orbit on both days. The predicted amplitudes of the $XCO_2$





enhancements are also comparable to those in the OCO-2 observation with small bias (less than 5 % for both days). As comparison, we also use the 2019 bottom-up $CO_2$ emissions to predict the $XCO_2$ enhancement on the two days (blue lines in Fig. 4 right panel). $XCO_2$ enhancements predicted by bottom-up $CO_2$ emissions deviate more from the OCO-2 observed enhancements than those predicted by our top-down $CO_2$ emissions. On 13 April 2020 in particular, the bottom-up

enhancement differs by +41% while the top-down differs by only within ±5 % compared to the observed $XCO_2$ enhancement. At the beginning of Wuhan's reopening, $CO_2$ emission from the city (our top-down estimation) is expected to be far lower than the pre-lockdown level (bottom-up estimation).

    We see that the estimated $XCO_2$ enhancement on 13 April 2020, both from the bottom-up and top-down emissions, are much 'narrower' compared to the OCO-2 observation. On this day, the OCO-2 orbit passes over the city center and the

dispersion plays an important role, which is neglected in the column model. In contrast, on 15 September 2018, the OCO-2 orbit passes downwind of the city center, and the width of the estimated and observed $XCO_2$ enhancements are more comparable. For comparison, we also conducted a Gaussian plume model to estimate $XCO_2$ enhancement (Text S4 and Fig. S6). On 13 April 2020, the result from Gaussian model agrees better with the OCO-2 observation, and on 15 September 2018, results from the two models (Gaussian model and the superposition column model) are close to each other and match

well with the observation.

    We also display $XCO_2$ enhancement line densities along wind direction with uncertainty on both days (Fig. 5). The line density shows a substantial increase of $XCO_2$ along the wind direction over the region with strong $CO_2$ emissions (Fig.5a, b, the inset maps). Where lines cross the OCO-2 orbit, the observed $XCO_2$ (as boxplots in Fig. 5a, b) are shown and their values agree with the predicted $XCO_2$ lines within ±0.3 ppm. It is remarkable that the $XCO_2$ enhancement is lower on 15 September

2018 than on 13 April 2020, despite $CO_2$ emissions on 15 September 2018 being nearly 65 % higher than those on 13 April 2020. The main reason for this is the lower wind speed on 13 April 2020, which accumulates pollutants over the city, and the fact that OCO-2 ground-track passed over the city center of Wuhan on this day. On 15 September, higher wind speeds and the OCO-2 track being situated over the outskirts of the city imply that a lower enhancement of $CO_2$ is observed.





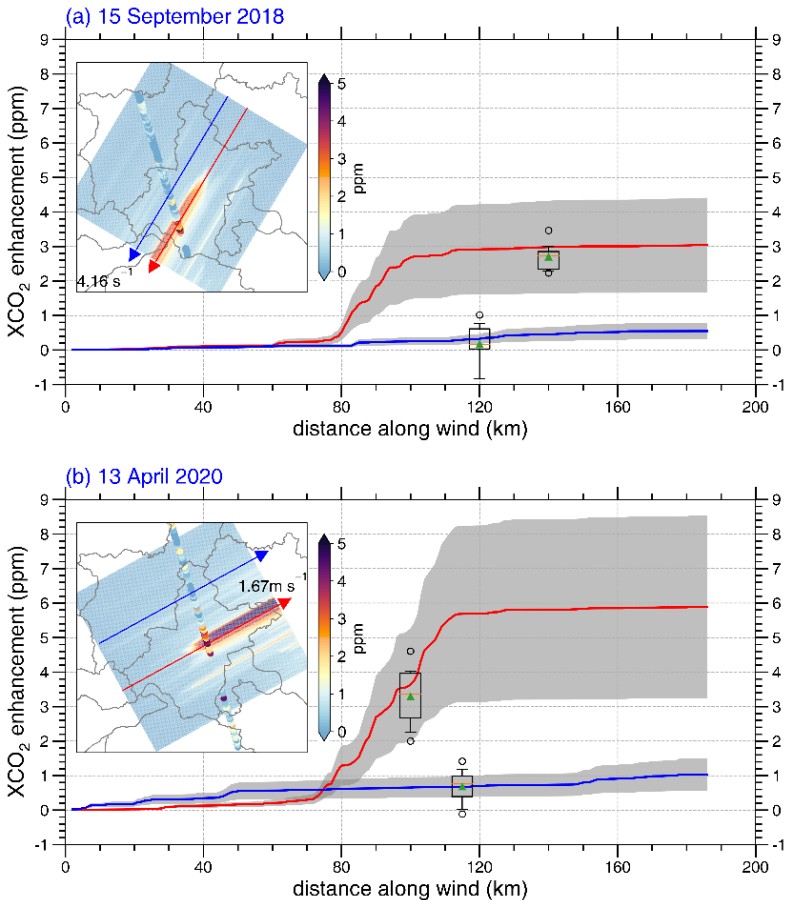

**Figure 5: Two presentative predicted XCO₂ enhancement lines (red and blue) on (a) 15 September 2018 and (b) 13 April 2020. When the XCO₂ enhancement line pass through the OCO-2 orbit, the observed XCO₂ enhancements are shown with boxplots, the mean values are shown as green triangles, the outliers beyond the 5–95 % interval are shown as circles. The predicted XCO₂ enhancement line density maps overlayed with OCO-2 observed XCO₂ enhancement on each day are shown inside, with the position of the presentative lines and the wind direction.**

We use an 'indirect' method to estimate daily city anthropogenic $CO_2$ emissions and $XCO_2$ enhancements, which may introduce uncertainties from the $NO_x$ emission estimation, the assumption of $CO_2$-to-$NO_x$ emission ratio, and the model to estimate $XCO_2$ enhancements. Despite all these uncertainties, we still generate daily Wuhan $CO_2$ emissions and $XCO_2$ enhancements that agree well with bottom-up inventory and OCO-2 observation, respectively.

### 3.3 Variation of $NO_x$ and $CO_2$ emissions in Wuhan from September 2019 to August 2020

Figure 6 displays the day-to-day variation of $NO_x$ and $CO_2$ emissions in Wuhan between September 2019 and August 2020. Before the pandemic of COVID-19, $NO_x$ emissions stay at a stable level of 11.53±1.08 kg s⁻¹, and $CO_2$ at 6.32±0.66 t s⁻¹, as





indicated by the dashed red lines. From January 2020 onwards, strict lockdown measurements were implemented to combat the COVID-19 pandemic, which led to lower industry production and less traffic on the road, and a sharp drop in $NO_x$ and $CO_2$ emissions (Ding et al., 2020; Zhang et al., 2020; Zheng et al., 2021b; Zhang et al., 2021; Feng et al., 2020). Our method

closely captures the timing and magnitude of these well-known sharp reductions in the emissions.

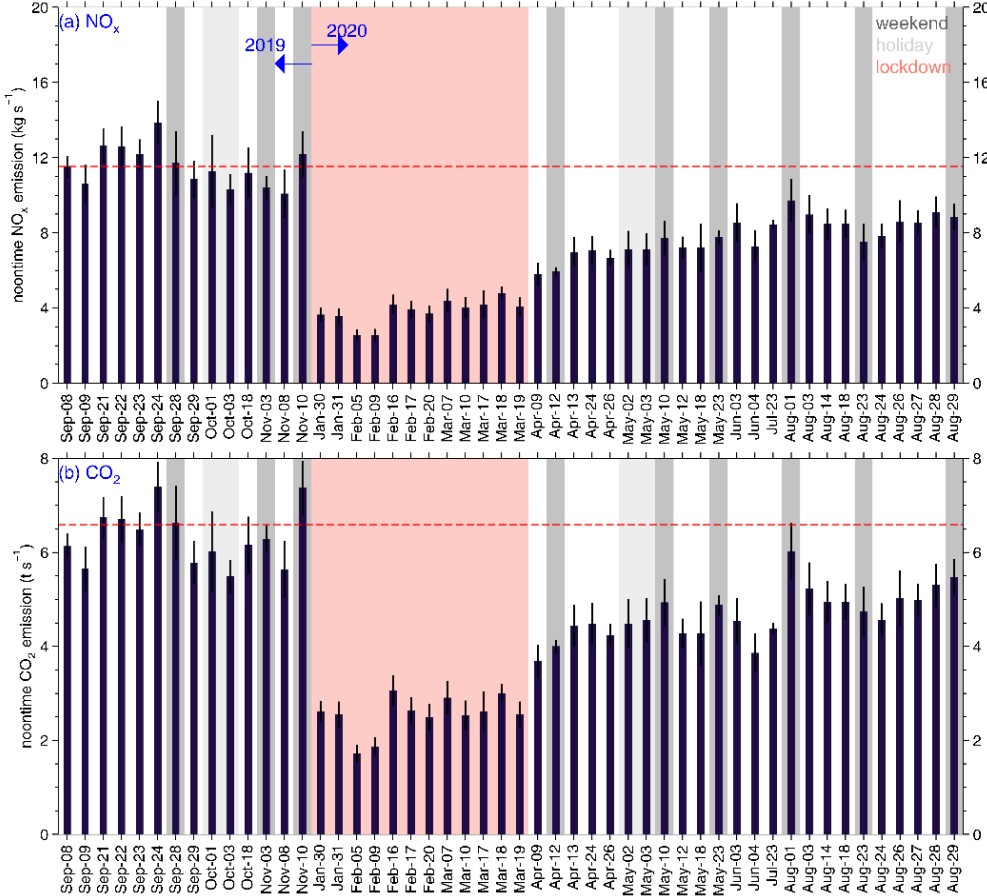

**Figure 6: 50 days (a) $NO_x$ and (b) $CO_2$ emissions in Wuhan estimated from TROPOMI from September 1 2019 to August 31 2020. The error bars denote the standard deviation of the five best estimates for each day, and the weekends, holidays and lockdown period are shaded with dark grey, light grey and red colors, respectively. The mean pre-lockdown emission levels are given as red**

**dashed lines.**

Wuhan $NO_x$ emissions on 30 January 2020 are $3.65\pm0.36$ kg s$^{-1}$, nearly 70 % lower than pre-lockdown levels, and decreased further and came to the lowest level in early February 2020, in accordance with Feng et al. (2020) who estimated similar reductions based on surface $NO_2$ observations. 5 February is the day with our lowest $NO_x$ emission from Wuhan of $2.55\pm0.31$ kg s$^{-1}$, only ~22 % of the normal level. $CO_2$ emissions have a similar temporal pattern as $NO_x$ emissions, but the

reduction relative to pre-lockdown level is smaller. The lowest $CO_2$ emission is at ~27 % of the pre-lockdown level (also on



5 February 2020), and the mean emission rate during the lockdown period (23 January to 8 April 2020) is 60 % lower than pre-lockdown level, while it is 67 % for $NO_x$. That $CO_2$ emission reductions are more modest than $NO_x$ reductions reflects the fact that the transportation sector had the strongest reductions during the lockdown, but since this sector also has the lowest $CO_2$-to-$NO_x$ ratios, the relative reduction in $CO_2$ remains somewhat smaller than in $NO_x$ emissions. This finding is
similar to that from Zheng et al. (2020b), who estimated the $NO_x$ and $CO_2$ emission variations for whole China.

From early February 2020 onwards, emissions increased slowly throughout the lockdown period. Wuhan $NO_x$ emission intensity in February 2020 was no more than 4.20 kg s$^{-1}$, some 60 % below the pre-lockdown level. Feng et al. (2020) estimated 61 % lower $NO_x$ emission from Wuhan in February 2020 than January based on surface $NO_x$ observations. Zheng et al. (2021a) reported a ~50 % lower $NO_x$ emission from Hubei in February 2020 than the annual mean level estimated from
a bottom-up approach.

Although Wuhan reopened on 9 April, the $NO_x$ and $CO_2$ emissions didn't see significant increases up until mid-May 2020. A perceptible increase in $NO_x$ emission is seen during late May, climbing to > 7.50 kg s$^{-1}$ ($NO_x$) and > 4.5 t s$^{-1}$ ($CO_2$), and leveling off thereafter. In August 2020, Wuhan $NO_x$ emissions were still some 25 % lower than the pre-lockdown level. Although bottom-up estimation by Zheng et al. (2021a) suggested that $NO_x$ emissions from the Hubei province were similar
in May–August 2020 as in 2019, surface and satellite observations over Wuhan show a 15–20 % lower $NO_2$ concentrations in May–August 2020 compared to 2019 (Fig. S7 and S8), consistent with our estimation of $NO_x$ emission. Liu et al. (2020c) reported 4.8 % higher $CO_2$ emissions for the whole China in August 2020 compared to August 2019. For the city of Wuhan, however, we calculate here some 20 % lower $CO_2$ emissions in August 2020 compared to the pre-lockdown level. Wuhan experienced a much more strict and longer period lockdown than other regions of China, and therefore a slower rebound of
$NO_x$ and $CO_2$ emissions should be expected over Wuhan.

**4 Conclusion**

In this study, we introduced an improved superposition column model to estimate daily $NO_x$ and $CO_2$ emissions from a Chinese megacity of Wuhan based on the latest released version 2.3.1 of TROPOMI $NO_2$ column data and OCO-2 $XCO_2$ observation. Our estimated daily $NO_x$ and $CO_2$ emissions agree well with bottom-up emissions with small bias of < 3 %.
Predicted $XCO_2$ enhancements based on our $CO_2$ emissions estimates prove to be in good agreement (within ±5 %) with OCO-2 observations over Wuhan. Compared to previous studies, our work shows that satellite measurements can provide detailed information on city-scale $NO_x$ and $CO_2$ emissions at unprecedented spatial and temporal resolutions. We achieved the day-to-day variation of $NO_x$ and $CO_2$ emissions from Wuhan between September 2019 and August 2020. We pointed out that the 'weekend reduction' is small, but that a 'holiday reduction' in Wuhan $NO_x$ and $CO_2$ emissions can be clearly
detected. We also captured the abrupt decrease in $NO_x$ and $CO_2$ emissions as the lockdown for COVID began on 23 January 2020, and the slow rebound as Wuhan reopened on 9 April 2020. Daily updates of city-scale $NO_x$ and $CO_2$ emissions provides policy makers with emission and policy control data on $NO_x$ and $CO_2$ emission control in urban environment.



In the future, following the launch of the Carbon Dioxide Monitoring mission (CO2M)(Sierk et al., 2021), our improved superposition column method may be explored further to constrain city scale $CO_2$ and $NO_x$ emissions to assess the effectiveness of emission control measures. CO2M provides simultaneous and co-located $CO_2$ and $NO_2$ observations with a wider swath than OCO-2, providing better opportunities to verify and improve $CO_2$ and $NO_x$ emissions from space.

.

**Author contributions.** Q.Z. and K.F.B designed the research; Q.Z. performed the data analysis, model development and result validation. B.Z. and H.Z. provide the ABACAS-EI $NO_x$ and $CO_2$ emission inventories. H.E. provides the 2.3.1 version of TROPOMI tropospheric $NO_2$ product. C.C. provides MEIC $NO_x$ emissions and perform the CMAQ simulations. X. Z. provided helpful discussions. Q.Z. and K.F.B. wrote the paper.

**Competing interests.** The authors declare no competing financial interest.

**Acknowledgement.** This work is funded by the National Natural Science Foundation of China (No.: 41805098) and the China Scholarship Council (202005330023). Improvements in TROPOMI $NO_2$ data have received support from the KNMI MSO NO₂NEXT project.

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
