# Peer review of "Quantifying daily NOx and CO2 emissions from Wuhan using satellite observations from TROPOMI and OCO-2"

_Atmospheric Chemistry and Physics, 2022_

## Author Comment (AC1)

**Reviewer #1:**

This paper studies the city scale  $NO_x$  and  $CO_2$  emissions on a very high spatial and temporal resolution. It provides insights into the real-time and detailed emission quantification and control of  $NO_x$  and  $CO_2$ . This work is interesting from a scientific point of view and is well organized and developed. Some minor revisions are suggested:

 Only the photochemical loss of NO2 is considered in the establishment of the superposition column model, how does the other pathways of NO2 loss? Are they also play a role in NOx chemistry?

Response: In the daytime, NOx are mainly subjected to photochemical reaction with the hydroxyl radical (OH) to produce nitric acid (HNO3), which is quickly converted to nitrate aerosols (NO3-). In some rural regions with substantial VOCs emissions, NOx may also react with VOCs :  $CH_3O_2 + NO_2 + M \leq PAN + M$  (e. g. Fischer et al., 2014). We have compared the GEOS-Chem model simulated HNO3+NO3- and PAN concentrations over our study domain in daytime:

Figure S3: GEOS-chem model simulated monthly mean HNO3+NO3- (left) and PAN (right) concentration for July 2020.

The modeled  $HNO_3+NO_3^-$  concentration is 5-10 times higher than PAN over Wuhan, indicating that  $NO_x$  loss via OH is the driving pathway of  $NO_x$  chemistry over our study domain in the daytime.

At night, NO2 is oxidized to HNO3 through the formation of N2O5 and NO3 and heterogeneous reactions including water vapor and aerosols (Shah et al., 2020; Lamsal et al., 2010). The overpass time of the satellite is 13:00–13:30 local time when NOx chemistry over the city is dominated by the photochemical process, so we consider the reaction between  $NO_x$  and OH in the superposition column model as the main loss pathway. We have clarified this point in the revised manuscript in Page 5, Line 133-134 and in the Text S2 and Fig. S3 in the revised supplementary material.

2. It is not clear to me how the 'starting background NO2 value' is determined.

Response: For each day, we calculate the mean NO2 line density value within the 5 (for summer, spring and autumn) or 10 (for winter) cells upwind of the starting cell of the study domain as the starting background NO2 value. Please refer to Page 5, Line 144-145 in the revised manuscript.

3. In line 140-145, the authors say that the negative  $\alpha$  value reflects the decay of upwind NO2 pollution along the wind, how come there are still positive  $\alpha$  values?

Response: In addition to the upwind NOx pollution, NOx emitted from natural sources like soil may also contribute to the background NO2 value. Soil NOx emissions are not influenced by wind direction, so on some days, when the background NO2 value is dominated by soil NOx rather than the upwind NOx, we may obtain a positive or zero value of  $\alpha$ .

4. The study obtains only 50 out of 365 days of valid data to quantify the  $NO_x$  and  $CO_2$  emissions, isn't it too few to estimate the daily variation of NOx and CO2 emissions?

Response: To assure the performance of the model, we filter out the days when cloud fraction is greater than 0.2 and the days when the wind direction shows substantial spatial or temporal variation within the study domain, and obtained 81 clear sky days with full satellite NO2 coverage within our study period. Then we excluded the days with fluctuating wind direction (if wind direction changes more than 45 degrees within 4 hours before TROPOMI overpass or over the study domain), which would lead to a less good correlation between modeled and observed NO2 line density (R less than 0.85). Finally, we obtain 50 out of the 365 days with reliable NOx and CO2 emissions estimation. The fraction of useful days is comparable to what Lorente et al. (2019) obtained for Paris, which is 27 days in 5 months. These 50 days covers at least 2 days for each month (except for December 2019). For 2019, it includes 9 workdays, 3 weekend days and 2

holiday days, which are enough to investigate the 'weekend reduction effect' and 'holiday reduction effect' in NOx emissions. It also covers 12 days across the lockdown period and 24 days after that, allowing us to monitor the large reduction and recovery of NOx and CO2 emissions from Wuhan due to the COVID lockdown. Therefore, these 50 days provide useful information to investigate the temporal emission patterns of NOx and CO2 from Wuhan and help to monitor the effectiveness of emission reductions in large urban centers. We have added this discussion in the revised manuscript in Page 16-17, Line 363-371.

5. Is there a difference in the overpass time of the TROPOMI and OCO-2 satellites? And how is this considered in the study?

Response: Both the TROPOMI and OCO-2 satellites overpasses at 13:00 – 13:30 local time, the difference is small, so we didn't consider this difference in the study.

6. According to Fig. S1, the predicted  $NO_x$  emission pattern is 'smoother' compared to the bottom-up emissions, do the authors think about the reason?

Response: We agree to the reviewer that the predicted NOx emission pattern is smoother compared to the bottom-up emissions. It is caused by two reasons. First, the spatial resolution of the bottom-up emission inventories is  $1 \text{ km} \times 1 \text{ km}$ , while it is  $5 \text{ km} \times 5 \text{ km}$ for the estimated emissions. Second, the spatial pattern of the bottom-up emission inventories is used as a first guess for the estimated emissions, but we let it shift along with the wind during the fitting, and the final predicted NOx emission pattern is determined by the mean of all the valid days. This is discussed further in the Section 1 of the revised supplementary material.

7. Fig. S4 shows that when the study domain is smaller, the estimated NOx lifetime is longer, how come?

Response: In the robustness test with respect to the area of study domain, we chose a smaller domain which encompasses the central part of Wuhan. The NO2 column density within this smaller domain is higher than outside. The OH radial is the major oxidizing agent to convert primary pollutants to secondary ones in the atmosphere, so the

concentration of OH radical concentration is lower inside because of titration (Tan et

al., 2018; Lorente et al., 2019). Consequently, the lifetime of  $NO_x$  inside the smaller domain will be longer. We have added this information in the revised supplementary material in Section 4.

Fischer, E. V., Jacob, D. J., Yantosca, R. M., Sulprizio, M. P., Millet, D. B., Mao, J., Paulot, F., Singh, H. B., Roiger, A., Ries, L., Talbot, R. W., Dzepina, K., and Pandey Deolal, S.: Atmospheric peroxyacetyl nitrate (PAN): a global budget and source attribution, Atmos. Chem. Phys., 14, 2679-2698, 10.5194/acp-14-2679-2014, 2014.

Lamsal, L. N., Martin, R. V., van Donkelaar, A., Celarier, E. A., Bucsela, E. J., Boersma, K. F., Dirksen, R., Luo, C., and Wang, Y.: Indirect validation of tropospheric nitrogen dioxide retrieved from the OMI satellite instrument: Insight into the seasonal variation of nitrogen oxides at northern midlatitudes, J. Geophys. Res., 115, 10.1029/2009jd013351, 2010.

Lorente, A., Boersma, K. F., Eskes, H. J., Veefkind, J. P., van Geffen, J., de Zeeuw, M. B., Denier van der Gon, H. A. C., Beirle, S., and Krol, M. C.: Quantification of nitrogen oxides emissions from build-up of pollution over Paris with TROPOMI, Sci. Rep., 9, 20033, 10.1038/s41598-019-56428-5, 2019.

Shah, V., Jacob, D. J., Li, K., Silvern, R. F., Zhai, S., Liu, M., Lin, J., and Zhang, Q.: Effect of changing  $NO_x$  lifetime on the seasonality and long-term trends of satellite-observed tropospheric  $NO_2$  columns over China, Atmos. Chem. and Phys., 20, 1483-1495, 10.5194/acp-20-1483-2020, 2020.

Tan, Z., Rohrer, F., Lu, K., Ma, X., Bohn, B., Broch, S., Dong, H., Fuchs, H., Gkatzelis, G. I., Hofzumahaus, A., Holland, F., Li, X., Liu, Y., Liu, Y., Novelli, A., Shao, M., Wang, H., Wu, Y., Zeng, L., Hu, M., Kiendler-Scharr, A., Wahner, A., and Zhang, Y.: Wintertime photochemistry in Beijing: observations of ROx radical concentrations in the North China Plain during the BEST-ONE campaign, Atmospheric Chemistry and Physics, 18, 12391-12411, 10.5194/acp-18-12391-2018, 2018.

---

## Author Comment (AC2)

**Reviewer #2:**

This study presents a new top-down superposition column model to estimate daily NOx and CO2 emissions from the largest city Wuhan in central China. It gives a very detailed description of this model, and the application to Wuhan clearly demonstrates the promising future. Overall, I think this manuscript is well structured, and the topic is suitable for ACP.

Major comments.

I think the authors need to clearly describe their way of calculating the uncertainty of their estimates. According to Section 2.6, large uncertainties are attached to the parameters used by the model (20% for OH, 10% of NOx/NO2, 30% for CO2-to-NOx, 15% of NOx lifetime). But the reported uncertainty from the text and Figure 3 is much lower (probably less than 10%). Could the authors specify how they calculate the full uncertainty? Have they used a Monte Carlo (or a similar) method to account for the uncertainty of each sub-process together? **Response: Thank you for this suggestion. We ran a test by randomly choosing parameter values within their uncertainty ranges for 20 times to predict an ensemble of NOx and CO2 emission outcomes is regarded as the uncertainty on NOx and CO2 emission caused by uncertainties in the corresponding parameters. The results in terms of uncertainties on NOx and CO2 emissions are listed below:**

| factor                                                | uncertainty | Influence on
NO x and (or) CO 2
emissions |
|-------------------------------------------------------|-------------|-----------------------------------------------------------------------|
| Satellite NO 2
retrieval                | ±20 %       | 20 %                                                                  |
| OH concentration                               | ±20 %       | 3 %                                                                   |
| NO 2 /NO x ratio                | ±10 %       | 8 %                                                                   |
| Wind field                                            | ±20 %       | 17 %                                                                  |
| CO 2 -to-NO x
emission ratio | ±30 %       | 30 %                                                                  |

The areas of the study domain may also lead to uncertainty in  $NO_x$  and  $CO_2$  emissions. We have narrowed down our study domain to check the sensitivity of our method to the chosen study domain (see Fig. S7). In the test, the study domain is narrowed down to 84 km in diameter, and, as expected, the result turns out to be structurally different from that with the 186 km diameter domain, for the mean OH concentration is lower in the city center, leading to longer fitted NOx lifetime. However, the change in fitted NOx lifetime and NOx emission is less than  $\pm 15\%$ . So we give an 15% uncertainty in NOx emission estimation caused by the size in the area of the study domain. All the uncertainty factors and their influence on NOx and CO2 emission estimation are listed in Table S2.

Finally, considering that all these parameters are independent from each other, we use the root mean square sum of the contributions to represent the overall uncertainty estimation, which we quantify for  $NO_x$  emission on a single day at ~31 %, and for  $CO_2$ emission at ~43 %. We updated our revised manuscript accordingly on Page 7-8, Line 192-202, Fig. 3 and Fig. 6 in the revised manuscript and Table S2 in the revised supplementary material.

Minor comments.

Line 21. Please specify the uncertainty. I believe the uncertainty of bottom-up inventories should be much larger than 3%.

Response: We agree. Differences between emissions from two inventories for the year 2019 were <3%, but the uncertainty is arguably larger than that. We rephrased the sentences: 'We estimated daily NOx and CO2 emissions from Wuhan between September 2019 to October 2020 with uncertainties of 31% and 43% respectively. Our estimated NOx and CO2 emissions are verified against bottom-up inventories with small mutual deviations (< 3 % for 2019 mean, ranging from -20 % to 48 % on a daily basis).'

Line 77. The retrieval methods can considerably affect the column concentrations. So it is better if the authors can comment on the related effects if using a newer version of TROPOMI data.

Response: This is exactly why we used two contemporary versions of the operational TROPOMI NO2 products (v1.3 and v2.3.1) and evaluated the impact of the retrieval version on our emission estimates. Improved (residual) cloud pressures correct the low bias of v1.3 data compared to OMI and ground-based measurements over east China

(Wang et al., 2020; Liu et al., 2020). In addition, an improved treatment for the surface albedo increases the columns for cloud-free scenes (Van Geffen et al., 2022). Compared to the earlier version, the v2.3.1 dataset has 10-40 % higher tropospheric NO2 columns over polluted scenes due to the improved cloud retrieval and other algorithm updates (Van Geffen et al., 2020; Riess et al., 2022). Over Wuhan we find an average increase in tropospheric NO2 of about 25%, but there are also differences between the two versions in terms of spatial and temporal distribution (Fig. S2). According to Fig. S2, the increase in v2.3.1 is much stronger over a polluted area (city center) and polluted period (9 September and 3 October 2019). The estimated  $NO_x$  lifetime and emissions from the two datasets for the whole study period are presented in Fig. S5. On average, using the **TROPOMI-v1.3** data leads to 13% lower NOx emissions from Wuhan than using the TROPOMI-v2.3.1 data. The NOx lifetime estimated from TROPOMI-v1.3 data is 5% shorter than that from TROPOMI-v2.3.1, which may be attributed to the fact that the TROPOMI-v2.3.1 data has a higher ratio between city center to the background. This information is added in the revised manuscript in Page 3 Line 79-91 and Page 9 Line226-231, and the revised Fig. S6.

---

## Author Response (AR1)

**Reviewer #1:**

This paper studies the city scale $NO_x$ and $CO_2$ emissions on a very high spatial and temporal resolution. It provides insights into the real-time and detailed emission quantification and control of $NO_x$ and $CO_2$. This work is interesting from a scientific point of view and is well organized and developed. Some minor revisions are suggested:

1. Only the photochemical loss of $NO_2$ is considered in the establishment of the superposition column model, how does the other pathways of NO2 loss? Are they also play a role in $NO_x$ chemistry?

**Response: In the daytime, $NO_x$ are mainly subjected to photochemical reaction with the hydroxyl radical (OH) to produce nitric acid ($HNO_3$), which is quickly converted to nitrate aerosols ($NO_3^-$). In some rural regions with substantial VOCs emissions, $NO_x$ may also react with VOCs : $CH_3O_2 + NO_2 + M <-> PAN + M$ (e. g. Fischer et al., 2014). We have compared the GEOS-Chem model simulated $HNO_3 + NO_3^-$ and PAN concentrations over our study domain in daytime:**

[Figure]

**Figure S3: GEOS-chem model simulated monthly mean HNO3+NO3- (left) and PAN (right) concentration for July 2020.**

**The modeled $HNO_3 + NO_3^-$ concentration is 5-10 times higher than PAN over Wuhan, indicating that $NO_x$ loss via OH is the driving pathway of $NO_x$ chemistry over our study domain in the daytime.**

**At night, $NO_2$ is oxidized to $HNO_3$ through the formation of $N_2O_5$ and $NO_3$ and heterogeneous reactions including water vapor and aerosols (Shah et al., 2020; Lamsal et al., 2010). The overpass time of the satellite is 13:00–13:30 local time when $NO_x$ chemistry over the city is dominated by the photochemical process, so we consider the**

**reaction between $NO_x$ and OH in the superposition column model as the main loss pathway. We have clarified this point in the revised manuscript in Page 5, Line 133-134 and in the Text S2 and Fig. S3 in the revised supplementary material.**

2. It is not clear to me how the 'starting background $NO_2$ value' is determined.

**Response: For each day, we calculate the mean $NO_2$ line density value within the 5 (for summer, spring and autumn) or 10 (for winter) cells upwind of the starting cell of the study domain as the starting background $NO_2$ value. Please refer to Page 5, Line 144-145 in the revised manuscript.**

3. In line 140-145, the authors say that the negative α value reflects the decay of upwind $NO_2$ pollution along the wind, how come there are still positive α values?

**Response: In addition to the upwind $NO_x$ pollution, $NO_x$ emitted from natural sources like soil may also contribute to the background $NO_2$ value. Soil $NO_x$ emissions are not influenced by wind direction, so on some days, when the background $NO_2$ value is dominated by soil $NO_x$ rather than the upwind $NO_x$, we may obtain a positive or zero value of α.**

4. The study obtains only 50 out of 365 days of valid data to quantify the $NO_x$ and $CO_2$ emissions, isn't it too few to estimate the daily variation of NOx and CO2 emissions?

**Response: To assure the performance of the model, we filter out the days when cloud fraction is greater than 0.2 and the days when the wind direction shows substantial spatial or temporal variation within the study domain, and obtained 81 clear sky days with full satellite $NO_2$ coverage within our study period. Then we excluded the days with fluctuating wind direction (if wind direction changes more than 45 degrees within 4 hours before TROPOMI overpass or over the study domain), which would lead to a less good correlation between modeled and observed $NO_2$ line density (R less than 0.85). Finally, we obtain 50 out of the 365 days with reliable $NO_x$ and $CO_2$ emissions estimation. The fraction of useful days is comparable to what Lorente et al. (2019) obtained for Paris, which is 27 days in 5 months. These 50 days covers at least 2 days for each month (except for December 2019). For 2019, it includes 9 workdays, 3 weekend days and 2**

**holiday days, which are enough to investigate the 'weekend reduction effect' and 'holiday reduction effect' in NO$_x$ emissions. It also covers 12 days across the lockdown period and 24 days after that, allowing us to monitor the large reduction and recovery of NO$_x$ and CO$_2$ emissions from Wuhan due to the COVID lockdown. Therefore, these 50 days provide useful information to investigate the temporal emission patterns of NO$_x$ and CO$_2$ from Wuhan and help to monitor the effectiveness of emission reductions in large urban centers. We have added this discussion in the revised manuscript in Page 16-17, Line 363-371.**

5. Is there a difference in the overpass time of the TROPOMI and OCO-2 satellites? And how is this considered in the study?

**Response: Both the TROPOMI and OCO-2 satellites overpasses at 13:00 – 13:30 local time, the difference is small, so we didn't consider this difference in the study.**

6. According to Fig. S1, the predicted NO$_x$ emission pattern is 'smoother' compared to the bottom-up emissions, do the authors think about the reason?

**Response: We agree to the reviewer that the predicted NO$_x$ emission pattern is smoother compared to the bottom-up emissions. It is caused by two reasons. First, the spatial resolution of the bottom-up emission inventories is 1 km × 1 km, while it is 5 km × 5 km for the estimated emissions. Second, the spatial pattern of the bottom-up emission inventories is used as a first guess for the estimated emissions, but we let it shift along with the wind during the fitting, and the final predicted NO$_x$ emission pattern is determined by the mean of all the valid days. This is discussed further in the Section 1 of the revised supplementary material.**

7. Fig. S4 shows that when the study domain is smaller, the estimated NOx lifetime is longer, how come?

**Response: In the robustness test with respect to the area of study domain, we chose a smaller domain which encompasses the central part of Wuhan. The NO$_2$ column density within this smaller domain is higher than outside. The OH radial is the major oxidizing agent to convert primary pollutants to secondary ones in the atmosphere, so the**

concentration of OH radical concentration is lower inside because of titration (Tan et al., 2018; Lorente et al., 2019). Consequently, the lifetime of $NO_x$ inside the smaller domain will be longer. We have added this information in the revised supplementary material in Section 4.

Fischer, E. V., Jacob, D. J., Yantosca, R. M., Sulprizio, M. P., Millet, D. B., Mao, J., Paulot, F., Singh, H. B., Roiger, A., Ries, L., Talbot, R. W., Dzepina, K., and Pandey Deolal, S.: Atmospheric peroxyacetyl nitrate (PAN): a global budget and source attribution, Atmos. Chem. Phys., 14, 2679-2698, 10.5194/acp-14-2679-2014, 2014.

Lamsal, L. N., Martin, R. V., van Donkelaar, A., Celarier, E. A., Bucsela, E. J., Boersma, K. F., Dirksen, R., Luo, C., and Wang, Y.: Indirect validation of tropospheric nitrogen dioxide retrieved from the OMI satellite instrument: Insight into the seasonal variation of nitrogen oxides at northern midlatitudes, J. Geophys. Res., 115, 10.1029/2009jd013351, 2010.

Lorente, A., Boersma, K. F., Eskes, H. J., Veefkind, J. P., van Geffen, J., de Zeeuw, M. B., Denier van der Gon, H. A. C., Beirle, S., and Krol, M. C.: Quantification of nitrogen oxides emissions from build-up of pollution over Paris with TROPOMI, Sci. Rep., 9, 20033, 10.1038/s41598-019-56428-5, 2019.

Shah, V., Jacob, D. J., Li, K., Silvern, R. F., Zhai, S., Liu, M., Lin, J., and Zhang, Q.: Effect of changing $NO_x$ lifetime on the seasonality and long-term trends of satellite-observed tropospheric $NO_2$ columns over China, Atmos. Chem. and Phys., 20, 1483-1495, 10.5194/acp-20-1483-2020, 2020.

Tan, Z., Rohrer, F., Lu, K., Ma, X., Bohn, B., Broch, S., Dong, H., Fuchs, H., Gkatzelis, G. I., Hofzumahaus, A., Holland, F., Li, X., Liu, Y., Liu, Y., Novelli, A., Shao, M., Wang, H., Wu, Y., Zeng, L., Hu, M., Kiendler-Scharr, A., Wahner, A., and Zhang, Y.: Wintertime photochemistry in Beijing: observations of $RO_x$ radical concentrations in the North China Plain during the BEST-ONE campaign, Atmospheric Chemistry and Physics, 18, 12391-12411, 10.5194/acp-18-12391-2018, 2018.

**Reviewer #2:**

This study presents a new top-down superposition column model to estimate daily NOx and CO2 emissions from the largest city Wuhan in central China. It gives a very detailed description of this model, and the application to Wuhan clearly demonstrates the promising future. Overall, I think this manuscript is well structured, and the topic is suitable for ACP.

Major comments.

I think the authors need to clearly describe their way of calculating the uncertainty of their estimates. According to Section 2.6, large uncertainties are attached to the parameters used by the model (20% for OH, 10% of NOx/NO2, 30% for CO2-to-NOx, 15% of NOx lifetime). But the reported uncertainty from the text and Figure 3 is much lower (probably less than 10%). Could the authors specify how they calculate the full uncertainty? Have they used a Monte Carlo (or a similar) method to account for the uncertainty of each sub-process together?

**Response: Thank you for this suggestion. We ran a test by randomly choosing parameter values within their uncertainty ranges for 20 times to predict an ensemble of $NO_x$ and $CO_2$ emission outcomes. Then the ratio of the standard deviation to the mean value of the 20 emission outcomes is regarded as the uncertainty on $NO_x$ and $CO_2$ emission caused by uncertainties in the corresponding parameters. The results in terms of uncertainties on $NO_x$ and $CO_2$ emissions are listed below:**

| factor | uncertainty | Influence on $NO_x$ and (or) $CO_2$ emissions |
|---|---|---|
| **Satellite $NO_2$ retrieval** | **±20 %** | **20 %** |
| **OH concentration** | **±20 %** | **3 %** |
| **$NO_2/NO_x$ ratio** | **±10 %** | **8 %** |
| **Wind field** | **±20 %** | **17 %** |
| **$CO_2$-to-$NO_x$ emission ratio** | **±30 %** | **30 %** |

**The areas of the study domain may also lead to uncertainty in $NO_x$ and $CO_2$ emissions. We have narrowed down our study domain to check the sensitivity of our method to the chosen study domain (see Fig. S7). In the test, the study domain is narrowed down to 84**

km in diameter, and, as expected, the result turns out to be structurally different from that with the 186 km diameter domain, for the mean OH concentration is lower in the city center, leading to longer fitted $NO_x$ lifetime. However, the change in fitted $NO_x$ lifetime and $NO_x$ emission is less than ±15%. So we give an 15% uncertainty in $NO_x$ emission estimation caused by the size in the area of the study domain. All the uncertainty factors and their influence on $NO_x$ and $CO_2$ emission estimation are listed in Table S2.

Finally, considering that all these parameters are independent from each other, we use the root mean square sum of the contributions to represent the overall uncertainty estimation, which we quantify for $NO_x$ emission on a single day at ~31 %, and for $CO_2$ emission at ~43 %. We updated our revised manuscript accordingly on Page 7-8, Line 192-202, Fig. 3 and Fig. 6 in the revised manuscript and Table S2 in the revised supplementary material.

Minor comments.

Line 21. Please specify the uncertainty. I believe the uncertainty of bottom-up inventories should be much larger than 3%.

Response: We agree. Differences between emissions from two inventories for the year 2019 were <3%, but the uncertainty is arguably larger than that. We rephrased the sentences: 'We estimated daily $NO_x$ and $CO_2$ emissions from Wuhan between September 2019 to October 2020 with uncertainties of 31% and 43% respectively. Our estimated $NO_x$ and $CO_2$ emissions are verified against bottom-up inventories with small mutual deviations (< 3 % for 2019 mean, ranging from -20 % to 48 % on a daily basis).'

Line 77. The retrieval methods can considerably affect the column concentrations. So it is better if the authors can comment on the related effects if using a newer version of TROPOMI data.

Response: This is exactly why we used two contemporary versions of the operational TROPOMI $NO_2$ products (v1.3 and v2.3.1) and evaluated the impact of the retrieval version on our emission estimates. Improved (residual) cloud pressures correct the low bias of v1.3 data compared to OMI and ground-based measurements over east China

(Wang et al., 2020; Liu et al., 2020). In addition, an improved treatment for the surface albedo increases the columns for cloud-free scenes (Van Geffen et al., 2022). Compared to the earlier version, the v2.3.1 dataset has 10–40 % higher tropospheric $NO_2$ columns over polluted scenes due to the improved cloud retrieval and other algorithm updates (Van Geffen et al., 2020; Riess et al., 2022). Over Wuhan we find an average increase in tropospheric $NO_2$ of about 25%, but there are also differences between the two versions in terms of spatial and temporal distribution (Fig. S2). According to Fig. S2, the increase in v2.3.1 is much stronger over a polluted area (city center) and polluted period (9 September and 3 October 2019). The estimated $NO_x$ lifetime and emissions from the two datasets for the whole study period are presented in Fig. S5. On average, using the TROPOMI-v1.3 data leads to 13% lower $NO_x$ emissions from Wuhan than using the TROPOMI-v2.3.1 data. The $NO_x$ lifetime estimated from TROPOMI-v1.3 data is 5% shorter than that from TROPOMI-v2.3.1, which may be attributed to the fact that the TROPOMI-v2.3.1 data has a higher ratio between city center to the background. This information is added in the revised manuscript in Page 3 Line 79-91 and Page 9 Line226-231, and the revised Fig. S6.

[Figure]

**Figure S6.** Estimated NO$_x$ (a) emissions and (b) lifetime over Wuhan during the study period based on the TROPOMI-v1.3 (blue bars) and TROPOMI-v2.3.1 (red bars) datasets. The error bars denote the corresponding uncertainty.

Line 88. Area-weighted sampling?

**Response: The satellite data is sampled in the regular longitude-latitude grid at a resolution of 0.05° (lon) × 0.05° (lat), and it is approximately 6 km × 6 km in the area of each grid. We didn't use an area-weighted sampling method, and a single pixel is only considered when its center falls in the cell. The NO$_2$ column density in each grid is calculated as the arithmetic mean of all the considered pixels.**

Line 105. Please specify the version of GEOS-Chem and give the reference. Also, give the full names of ERA5 and ECMWF

**Response: added. We use the version 12.1 of GEOS-Chem model with a horizontal resolution of 0.25° × 0.3125° (~ 30 × 37.5 km$^2$) to provide the a priori guesses for chemical parameters relevant to daytime NO$_x$. The wind field is from ERA5 (ECMWF Reanalysis v5), the fifth generation ECMWF (European Centre for Medium-Range Weather Forecasts) atmospheric reanalysis of the global climate.**

Line 148. The correlation coefficient of pixels along the wind direction is not a very useful metric here because these pixels are not independent of each other. It is better to show the mean bias along with the correlation coefficients.

**Response: The reviewer's point is well-taken. The mean bias is now also included in the revised manuscript in Page 6 Line 158-161, and is displayed in the revised Fig. 1. Both the correlation coefficient and the mean bias are useful metrics to evaluate the performance of the superposition model. The correlation coefficient quantifies the success with which the fitting model reproduces the observed line densities, and the mean bias describes the deviation of fitted line densities from the observations. Compared to fitting results with a constant background value, we obtain a better correlation (up to 25%) and lower bias (nearly 50% lower) between fitted and observed NO$_2$ line densities then when fitting with a linearly changing background value.**

Line 342. Please specify the resolutions here.

**Response: This sentence is rephrased as "Compared to previous studies, our work shows that satellite measurements can provide detailed information on sub-city scale NO$_x$ and CO$_2$ emissions on daily basis".**

Liu, M., Lin, J., Kong, H., Boersma, K. F., Eskes, H., Kanaya, Y., He, Q., Tian, X., Qin, K., Xie, P., Spurr, R., Ni, R., Yan, Y., Weng, H., and Wang, J.: A new TROPOMI product for tropospheric NO$_x$ columns over East Asia with explicit aerosol corrections, Atmos. Meas. Tech., 13, 4247-4259, 10.5194/amt-13-4247-2020, 2020.

Riess, T. C. V. W., Boersma, K. F., van Vliet, J., Peters, W., Sneep, M., Eskes, H., and van Geffen, J.: Improved monitoring of shipping NO$_2$ with TROPOMI: decreasing NO$_x$ emissions in European seas during the COVID-19 pandemic, Atmos. Meas. Tech., 15, 1415-1438, 10.5194/amt-15-1415-2022, 2022.

van Geffen, J., Boersma, K. F., Eskes, H., Sneep, M., ter Linden, M., Zara, M., and Veefkind, J. P.: S5P TROPOMI NO$_2$ slant column retrieval: method, stability, uncertainties and comparisons with OMI, Atmos. Meas. Tech., 13, 1315-1335, 10.5194/amt-13-1315-2020, 2020.

van Geffen, J., Eskes, H., Compernolle, S., Pinardi, G., Verhoelst, T., Lambert, J.-C., Sneep, M., ter Linden, M., Ludewig, A., Boersma, K. F., and Veefkind, J. P.: Sentinel-5P TROPOMI NO$_2$ retrieval: impact of version v2.2 improvements and comparisons with OMI and ground-based data, Atmos. Meas. Tech., 15, 2037-2060, 10.5194/amt-15-2037-2022, 2022.

Wang, C., Wang, T., Wang, P., and Rakitin, V.: Comparison and Validation of TROPOMI and OMI NO2 Observations over China, Atmosphere, 11, 10.3390/atmos11060636, 2020.